# Improvement of Ecological Risk Considering Heavy Metal in Soil and Groundwater Surrounding Electroplating Factories

**Hong Fang** [1,†], **Xiujuan Wang** [1,†], **Di Xia** [1], **Jianting Zhu** [2], **Weida Yu** [1], **Yaoming Su** [1], **Jingwen Zeng** [1], **Yuanling Zhang** [1], **Xiaojun Lin** [1], **Yutao Lei** [1,*] and **Jinrong Qiu** [1,*]

1 South China Institute of Environmental Sciences, Ministry of Environmental Protection of China, Guangzhou 510655, China; fanghong@scies.org (H.F.); wangxiujuan2009@163.com (X.W.); xiadi@scies.org (D.X.); yuweida116305234@163.com (W.Y.); suyaoming@scies.org (Y.S.); zengjingwen@scies.org (J.Z.); z15521129530@163.com (Y.Z.); linxiaojun@scies.org (X.L.)
2 Department of Civil and Architectural Engineering, University of Wyoming, Laramie, WY 82071, USA; jzhu5@uwyo.edu
* Correspondence: leiyutao@scies.org (Y.L.); qiujinrong@scies.org (J.Q.)
† These authors contributed equally to this work.

**Abstract:** Heavy metals in groundwater and soil are toxic to humans. An accurate risk assessment of heavy metal contamination can aid in environmental security decision making. In this study, the improved ecological risk index (*RI*) is used to comprehensively investigate the influence of heavy metals in soil and groundwater within electroplating factories and their surrounding regions. In the non-overlapping area, the *RI* of soil and groundwater is computed individually, and in the overlapping area, the greater *RI* of soil and groundwater is employed. Two typical electroplating factories are used to examine the heavy metal distribution pattern. The heavy metal concentrations are compared between Factory A, which is in operation, and Factory B, which is no longer in operation, in order to analyze the heavy metal concentrations and associated ecological risks. Heavy metals continue to spread horizontally and vertically after Factory B was closed. Heavy metal concentrations in groundwater surrounding Factory B are substantially greater, and the maximum concentration exists deeper than in Factory A. Because Cr, Cu, and Hg in soil contribute significantly to the *RI*, the primary high *RI* region is observed at Factory A and the region to the southwest. The *RI* of Factory B demonstrates a broad, moderate risk zone in the west and southwest.

**Keywords:** heavy metal; electroplating factory; groundwater; soil; multivariable analysis; environmental evaluation



## 1. Introduction

Contamination of the environment caused by direct disposal of heavy metals has become a serious issue in recent years. High quantities of Cu, Ni, Cr, and Zn are commonly found in electroplating factory waste [1–3]. Long-term irrigation with electroplating effluent may result in incremental heavy metal poisoning of soil and groundwater [4]. Although soils are efficient filters for adsorbing and maintaining heavy metals, when their retention capacity falls, soils release heavy metals into the groundwater. Precipitation makes it easier for surface water to transport heavy metals to groundwater via infiltration. Heavy metals may form insoluble complex compounds with organic substances in the soil [5]. The majority of contaminants are non-biodegradable and non-thermodegradable, which can result in natural and environmental devastation, damage, and disastrous outcomes. Furthermore, heavy metals slowly and unnoticeably accumulate to dangerous levels that endanger animal and plant life [6,7]. Heavy metals in contaminated soil and groundwater can be ingested, inhaled, and come in contact with the human body directly or indirectly [8,9]. Previous studies' ecological risk assessments solely examined the effect of heavy metals in relatively stable soil. Actually, polluted groundwater has a considerable impact on

ecological risk assessment, because heavy metals diffuse far quicker in groundwater than they do in soil. By only considering the effects of heavy metals in soil, the ecological risk assessment is greatly underestimated.

Hundreds of electroplating facilities have been built in Guangzhou, South China, due to the steady and fast expansion of the economy and technology. The polluted spatial pattern in these electroplating factories is typically the same. Guangzhou is controlled by the East Asian monsoon's seasonal variations, which provide a lot of rain in the summer. Because of the plentiful precipitation and shallow groundwater table, heavy metals discharged from electroplating facilities can readily pollute the soil and groundwater. Previous research investigated the dangers of heavy metal to human health as well as environmental remedies. Wang et al. [10] analyzed the sediments near the Shekou Industrial District's exit and discovered that zinc (Zn), copper (Cu), cadmium (Cd), and mercury (Hg) were mostly derived from effluent from the electroplating, metal, and battery industries. Using polluted wastewater for irrigation, Xiao et al. [11] investigated the distribution of heavy metals in the vegetable and paddy crops near an electroplating facility. Zeng et al. [12] investigated the distribution of heavy metals and built a model for determining the possible damage to humans. The cancer risk was around 20% higher in the region with hazardous heavy metals in the drinking water than in the rest of the country [13]. Those studies, on the other hand, do not precisely identify the source of heavy metal pollution or explain how it occurs.

The distribution of heavy metals in soil and groundwater is influenced by soil factors such as soil type, pH, and land use. In order to improve soil and groundwater reclamation, it is necessary to examine metal behavior [14,15]. The use of multivariate statistical analysis in conjunction with geostatistical techniques can provide precise access to heavy metal behavior in soil and groundwater, identify contamination sources, investigate spatial distribution, assess risk at a specific site, and develop future reclamation projects in industrial regions [16–19].

The contaminated soil and groundwater containing heavy metals near electroplating facilities in Guangzhou, China, are thoroughly researched in this study. The primary goal of this research is to investigate the interaction of heavy metals such as Cd, Cu, Cr, and Zn with soil and groundwater in electroplating facilities and their surrounding areas. The degree of metal pollution and ecological risks to the environment are evaluated by comparing the variation between a typical electroplating facility in production and another that was shut down three years ago. The comparison can help to identify the source of heavy metal pollution near electroplating plants and to comprehend the process of heavy metal contamination. By examining the distribution of the soil and groundwater profile, we may also identify high-risk locations where heavy metals might readily concentrate. The revised technique of calculating the *RI* can be applied to any electroplating factory to determine the possible environmental risk.

Groundwater contamination is heavily influenced by groundwater flow direction and is mostly disseminated in the groundwater's downstream area. The area contaminated by soil is heavily influenced by wind direction. These two areas are not interchangeable. Most previous studies calculated the *RI* of soil or groundwater individually and did not account for the influence of both. This study is unique in that it comprehensively evaluates both soil and groundwater contamination in and around the electroplating factory. In the non-overlapping area, the *RI* of soil and groundwater is computed individually, and in the overlapping area, the greater *RI* of soil and groundwater is employed. The soil and groundwater risk indices were thoroughly compared, taking into account the continuous discharge of pollution sources from the electroplating facilities and the cessation of discharge of pollution sources following the closure of the electroplating factory.

## 2. Materials and Methods

### 2.1. Study Areas

The study areas, which are located northwest of Guangzhou and have been polluted by human activities, particularly heavy metals, have produced major environmental is-

sues. The research locations are two particular areas surrounding two electroplating plants (Factory A and Factory B, which are 16 km apart) in Guangzhou, as illustrated in Figure 1. A subtropical monsoon climate dominates these two places, with an average annual precipitation of 1694 mm and a mean annual temperature of roughly 21.8 °C [20]. Its climate is distinguished by hot summers and mild winters. Precipitation occurs frequently and quickly recharges the groundwater and subsurface water systems. The groundwater has a shallow water table that is only 0.8 m below the surface.

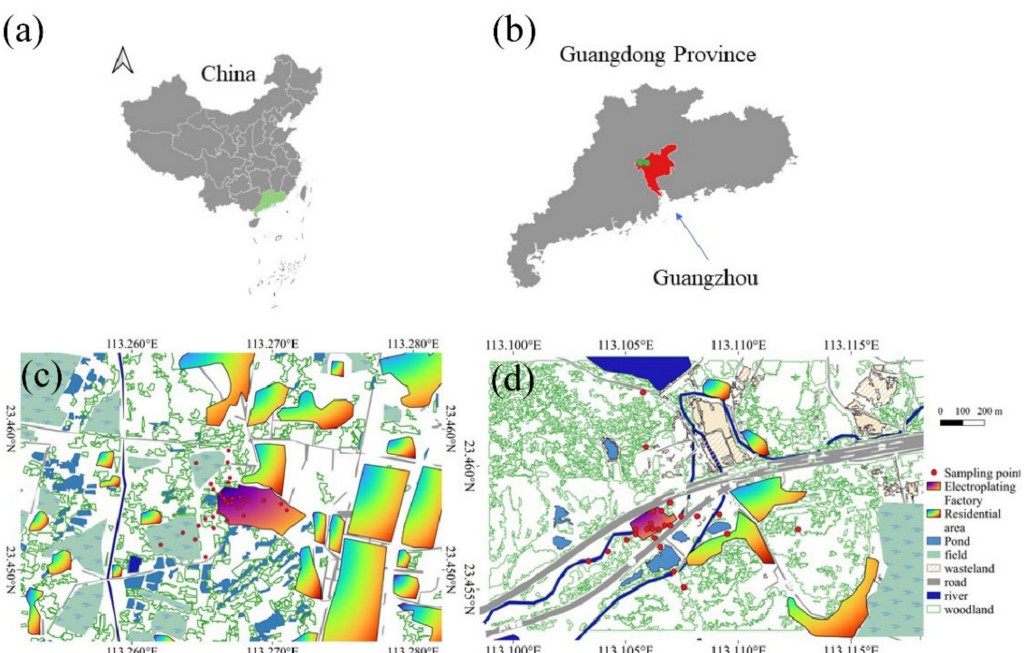

**Figure 1.** (**a**) Location of Guangdong province in China. (**b**) Location of Guangzhou in Guangdong province. (**c**) Map of Factory A and sampling sites. (**d**) Map of Factory B and sampling sites.

Factory A was constructed in 2000 and its primary manufacturing job is to apply a layer of chromium metal coating to the surface of plated components, giving the surface of the plated parts a certain wear resistance and a bright silver-white appearance. About 10 million electroplating parts are produced each year. The effluent from the factory, currently, largely contains pollutants such as Cr, Cr (VI), and Ni. Factory B was established in 1986 and its contaminated source is primarily electroplating wastewater. It produces about 9 million electroplating parts per year. Throughout the electroplating production process, the workpiece must be washed from one operation to the next. Heavy metals such as Ni and Cr are commonly found in electroplating wastewater discharged by the factory. The surroundings of these two electroplating factories are similar, with the primary terrain types being residential areas, farmland, and bare ground. Factory A is still in operation, whereas Factory B has been inactive for three years. Both factories are situated on a relatively flat area of land. High quantities of Cr (VI), Cr, Cu, Ni, Zn, and Hg have been discovered in electroplating wastewater and soil around Factory A. The residential areas are located in the north, northeast, east, and southeast, whereas the rest of the land is farmland. Cr (VI), Cu, Ni, Zn, and Cr concentrations are high in the soil and groundwater at Factory B. The residential neighborhood is mostly located in the east and northeast at a distance of 150 m. Farmland can be found in the west, north, and south areas, whereas grass can be seen in the south.

### 2.2. Sample Collection and Preparation

The sample point sites were selected to follow the radial directions with dense distribution around the factory, as illustrated in Figure 1c,d for the two factories, respectively, with the electroplating area and sewage treatment area as the center. To eliminate accidental

measurement errors, soil samples were collected and separated into four portions. A total of 1.5 kg of heavy metal samples were collected and sorted into polythene zip-lock bags. After collecting the soil samples, they were placed in a chilled box and transported to the laboratory for testing within 24 h. Both soil samples were taken from the two places in varied soil levels. For Factories A and B, 40 and 33 soil samples, as well as 6 and 5 groundwater samples, were obtained, respectively. The groundwater samples were collected in 100 mL prewashed narrow-mouth polyethylene bottles. The samples were filtered using 0.45 m filter paper, and they were preserved using ultrapure 6 N nitric acid (pH 2).

### 2.3. Chemical Analysis

To begin, we digested 0.1 g of each sample with an 8 mL mixed acid solution of $HNO_3$, $HClO_4$, and HF at a volume ratio of 5:4:1. The mixture was heated at 120 °C for 12 h to thoroughly digest the heavy metals in the sediment [21]. The mixture was then heated on a hot plate until it was completely dry. The material was then diluted with double distilled water and filtered through a 0.45 m membrane. The filtrate was then transferred to a centrifuge tube for examination. Inductively coupled plasma mass spectrometry (ICP-MS) and inductively coupled plasma atomic emission spectrometry (ICP-AES) were used to measure the concentrations of six elements (Cd, Cr, Hg, Cu, Zn, and Ni) [22,23]. To assure the accuracy of the analyses, the standards of international quality assurance and control procedures need to be satisfied. These approaches are presented in detail by Chen et al. [24] and Wu et al. [25].

### 2.4. Interpolation Method and Environmental Assessment Indicators

The accuracy of heavy metal distribution in soil and groundwater is affected by the interpolation method adopted. The inverse distance weighted (IDW) method is suited for survey data that is spread at irregular geographical intervals [26]. The IDW method was used in this study to characterize the contour of the pollutant concentration. The prospective ecological risk index (*RI*) is commonly used to assess the ecological risk of a variety of heavy metals in soil and groundwater [27]. The estimated findings can provide insight into the risk assessment of heavy metal contamination and aid in environmental security decision making. The improved *RI* value is calculated as follows:

$$C_{fg}^i = \frac{C_g}{B_g} \tag{1}$$

$$C_{fs}^i = \frac{C_s}{B_s} \tag{2}$$

$$E_{rg}^i = T_{rg}^i \cdot C_{fg}^i \tag{3}$$

$$E_{rs}^i = T_{rs}^i \cdot C_{fs}^i \tag{4}$$

$$RI = \sum_{i=1}^n Max\left(E_{rg}^i, \ E_{rs}^i\right) \tag{5}$$

where *B* means the background concentration; *C* denotes the heavy metal concentration; and *T* denotes the toxicity response factor of each heavy metal. Cr (VI) = 60, Cr = 2, Zn = 1, Hg = 40, Cu = 5, and Cd = 30 were the values employed in this investigation [28,29]. The toxicity response factor signifies the contamination factor as well as the heavy metal's potential harm. Groundwater and soil are represented by the subscripts g and s, respectively. The integrated ecological risk for various heavy metals is referred to as *RI*. Table 1 shows the grading requirements for possible ecological risk [30].

**Table 1.** Grading standards of potential ecological risk.

| *RI* | **Grades of Potential Ecological Risk to the Environment** |
|---|---|
| *RI* < 150 | Low risk |
| 150 ≤ *RI* < 300 | Moderate risk |
| 300 ≤ *RI* < 600 | Considerable risk |
| *RI* ≥ 600 | High risk |

*2.5. Statistical Analysis*

In environmental research, multivariate statistical techniques are commonly employed [31]. To investigate heavy metal concentrations, we employed Pearson's correlation analysis, principal component analysis (PCA), and cluster analysis (CA) in this work. Pearson's correlation analysis can be used to arrange enormous data sets in order to understand the relationships between variables [32]. It was used to examine the links between heavy metals and soil features, as well as groundwater. In multivariate problems, PCA is utilized as a mathematical tool using an unsupervised, linear pattern recognition approach for categorizing and decreasing the dimensionality of numerical data sets [33]. It was used to explore the factors that influence the variance of total heavy metal concentration. CA is used to classify heavy metals and build a group of comparable clusters [34]. In this study, CA was used to evaluate the variations in the influence of each heavy metal.

**3. Results**

*3.1. The Distribution of Heavy Meal in Soil and Groundwater*

Figure 2 depicts the regional pattern of heavy metal concentrations in Factory A's soil and groundwater. Figure 2a–f indicate that the concentrations of Cr (VI), Cu, Ni, Zn, Cr, and Hg within the industrial site are greater than in the natural soils around the factory, indicating that the electroplating factory is the primary source of pollution. The five heavy metals, Cr (VI), Cu, Ni, Zn, and Cr, show comparable geographical concentration patterns in which pollutants migrate mostly to the northeast, implying that these five heavy metals are likely to have similar origins. Figure 2e shows that there is most likely another pollution discharge outlet to the southwest of the plant site, particularly for Cr. Meanwhile, the element Hg has similar regional patterns, with larger concentrations within the production site. In comparison to the other five heavy metals, the distribution of Hg is substantially smaller.

The contours of heavy metal content in groundwater are depicted in Figure 2g–l. The elements have similar geographical distributions, with the maximum concentration on the manufacturing site and decreasing gradually away from the factory. However, the concentration gradient is more visible along the north–south axis than along the east–west axis. The fundamental cause might be that the aquifer's conductivity is greater in the north–south direction than in the east–west direction. Furthermore, the comparable distribution indicates that these six elements are most likely derived from the same source invading the groundwater.

Figure 3a–e depict the concentration distribution patterns of Cr (VI), Cu, Ni, Zn, and Cr, with larger concentrations at the industrial site. The same distribution indicates that these five heavy metals originated from the same sources. The concentration gradient of these five heavy metals extends mostly northwest–southeast. Despite the fact that the factory was closed three years ago, the high concentration remains, indicating long-term pollution in the soil. Figure 3f–j depict a similar pattern of the five heavy metal pollutions of groundwater at concentrations far beyond natural environmental value.

Soil profiles taken in electroplating sites are used to study the concentration of heavy metals from the ground's surface to the groundwater table. The patterns of heavy metal concentrations with depth are compared in Factory A and B. Figure 4 depicts the concentrations of Cr (VI), Cu, Ni, Zn, Cr, and Hg in proportion to depth, with the surface having the maximum concentration. For Factories A and B, Zn and Cu concentrations are at their

maximum level in the surface soil at a depth of 0–20 cm. Surface soils, on the other hand, have less Cu, Cd, and Cr accumulation, but groundwater has a high concentration of Cu, Cd, and Cr, showing that heavy metals in soils can permeate into groundwater. The average depth of Cr (VI) contamination in Factory A is around 300 cm. The concentration of Cr (VI) in Factory B is substantially lower than in Factory A. Under acidic circumstances, Cr (VI) migrates more easily from soil to groundwater. This might explain why the majority of Cr (VI) in the closed Factory B is transferred to groundwater. Figure 4d reveals that the visible Cu concentration in both factories is mostly detected in the 0–400 cm and 0–600 cm ranges, respectively. Cu in Factory B reaches greater depths than Cu in Factory A. The majority of Cu occurs as stable organic complexes and is adsorbed on the surface of soil colloids. Because Cu has a limited solubility and mobility, the Cu content in Factory B remained high after closure.

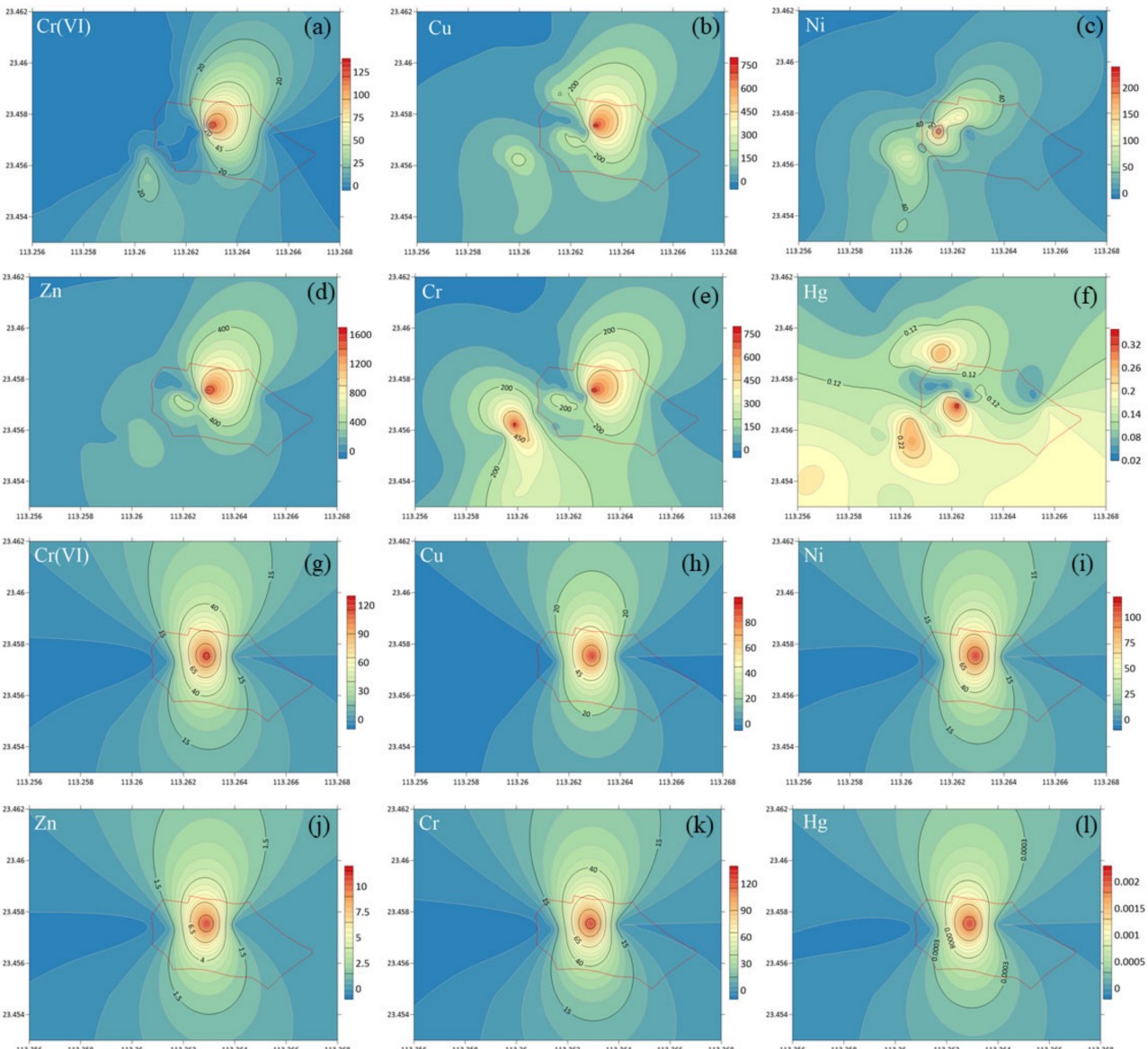

**Figure 2.** Distribution of heavy metals in surface soil (0–20 cm) of Factory A and its surrounding area. (**a**–**f**) are the contour plots of the heavy metal concentration (mg/kg) in soil. (**g**–**l**) are the contour plots of the heavy metal concentration (mg/L) in groundwater.

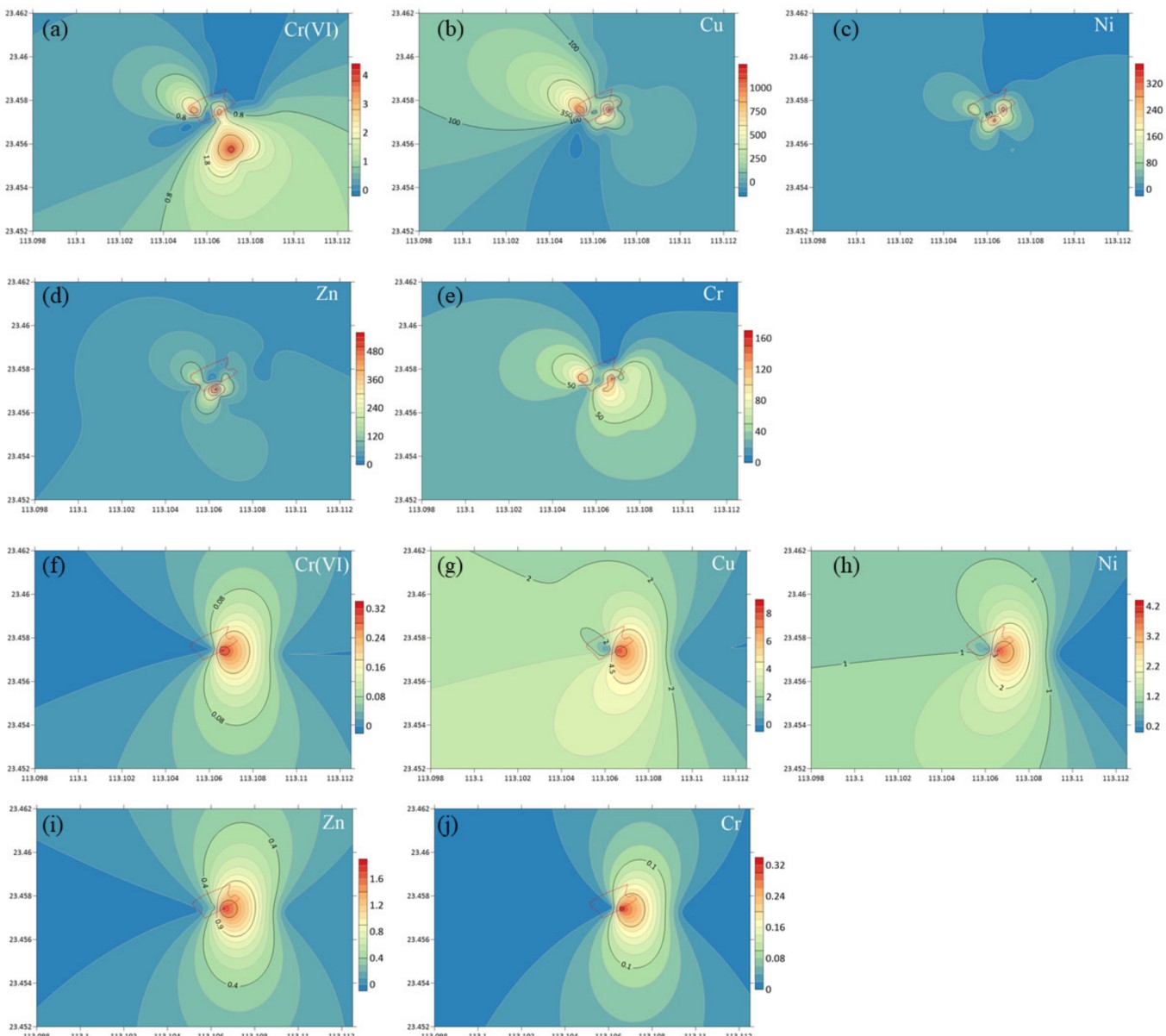

**Figure 3.** Distribution of heavy metals in surface soil (0–20 cm) of Factory B and its surrounding area. (**a**–**e**) are the contour plots of the heavy metal concentration (mg/kg) in soil. (**f**–**j**) are the contour plots of the heavy metal concentration (mg/L) in groundwater.

As shown in Figure 4a–f, similar heavy metal concentration patterns occur for both industries, with heavy metal concentrations being greatest in the surface soil, within the range of 0 to 20 cm. Cu, Ni, Zn, and Cr in soil exhibit a similar trend in both factories, rapidly reducing with depth, especially around 30 cm, and gradually decreasing at a higher depth, suggesting that the diffusing velocity is fast in the topsoil but dramatically lowers in deeper soil. The greatest concentrations of Zn and Cu in surface soil are found in Factory A and Factory B, respectively (Figure 4d,e). It follows that the Zn and Cu concentrations in groundwater are likewise the greatest (Figure 4j,k). In contrast, both Cr (VI) concentrations are lowest in groundwater because Cr (VI) concentrations are lowest in soil. The concentration of Hg in soil and groundwater follows a similar pattern, with the maximum concentration in the surface soil. The link between heavy metal concentrations in soil and groundwater shows that soil transfers heavy metals into groundwater. In comparison to Factory B, the greatest heavy metal concentrations in Factory A are found

mostly in shallower groundwater. The largest concentrations of Cr (VI), Zn, and Cr are found at roughly 300 cm, showing that the heavy metal enters by gravity and leaching.

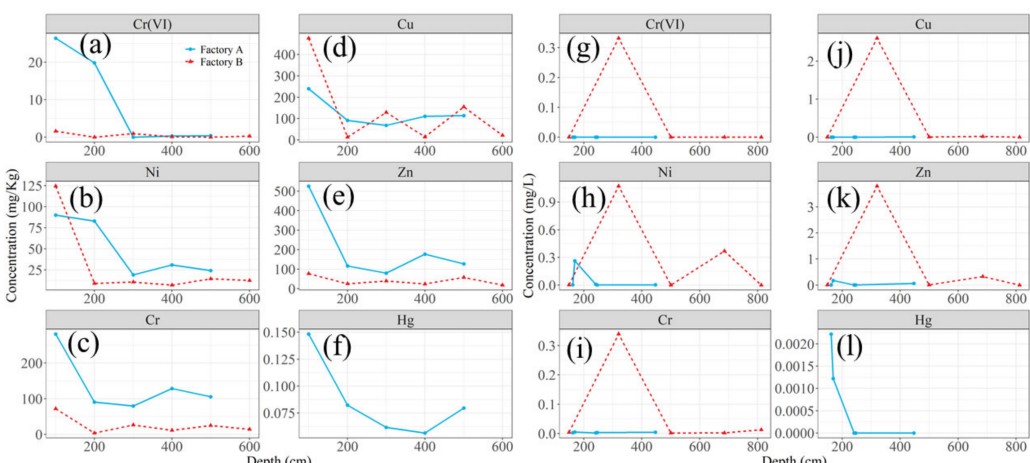

**Figure 4.** Concentrations of Cr (VI), Cu, Ni, Zn, Cr, Hg in the soil (**a**–**f**) and groundwater (**g**–**l**) in relation to the depth. The solid and the dashed lines denote the concentrations in Factory A and B, respectively. The highest concentration of the heavy metal is observed in the topsoil for both factories (**a**–**f**). Most of the maximum concentration is observed in shallower places for Factory A whereas the maximum concentration is observed at deeper places for Factory B in groundwater (**g**–**l**).

### *3.2. Results of Multivariable Analysis*

3.2.1. Impact of Factory Site and Its Surrounding Regions Using Primary Component Analysis (PCA)

Figure 5 depicts each heavy metal's contribution to the major component loading from the factories. Figure 5a displays the two important principle components and their corresponding percentages of total variance for Factory A, which directly represents the link between the two principal components depending on the degree generated by heavy metals. The four heavy elements Cr (VI), Cu, Cr, and Zn contributed 55.6 percent of the total variance's first main component (PC1) and had strong correlations (loading > 0.70). The second, principal component 2 (PC2), had the highest weight of Hg and provided 21.2 percent of the overall variance. PC1 and PC2 accounted for 76.8 percent of the total, implying that the two PCs represent geochemical variability. The ellipses depict the areas of the factory site and its surrounding area, which are grouped by a 68 percent confidence interval. The ellipse representing the territory inside the factory is significantly bigger than the ellipse representing the region outside the factory, indicating that the effect caused by heavy metal pollution inside the production site is much greater than the area outside the facility.

Figure 5b depicts the PCA findings for Factory B. PC1 accounted for 67.1 percent of overall variation and had high loading values for Cu, Cr (VI), Cr, Ni, and Zn. PC2 contained heavy metals such as Cr and Ni, which accounted for 21.4 percent of the variance. As a result, PC1 and PC2 represent components with distinct origins in plating procedures. In this scenario, the samples had much greater Cu contents, which might be attributed to actions during the electroplating operation. The largest quantities of Cr (VI), Cu, Ni, Zn, and Cr were found in the production workplace, explaining the severe metal pollution. The angle formed by the ellipse's main axis and the positive direction of the horizontal axes inside and outside the factory are approximately 20° and 120°, respectively. This demonstrates how heavy metal ions in the factory's soil are carried to the region outside the factory via groundwater. Concentration falls within the factory while increasing outside the factory. The concentration between these two areas is inversely connected. The areas showing a 68 percent confidence interval eclipse of these two locations are almost identical, suggesting that the impact of contamination of the regions is imminent.

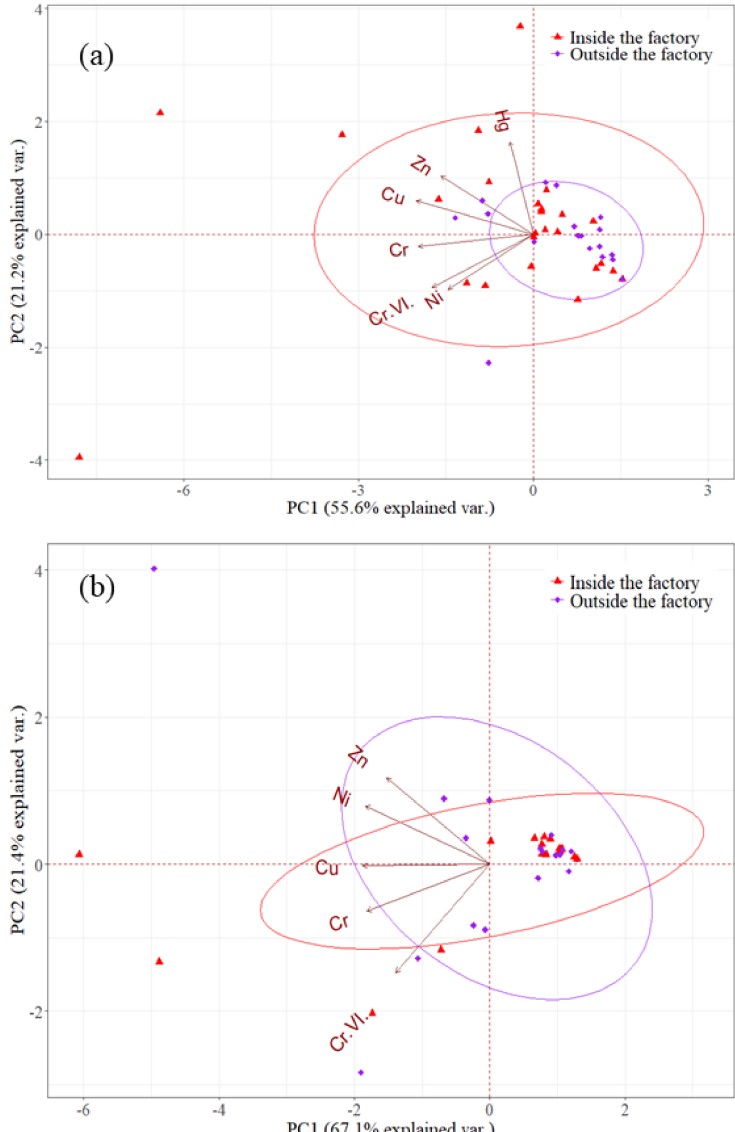

**Figure 5.** Contribution of each heavy metal to the primary component loading obtained by the principal component analysis from factories. The triangle points denote the samples collected in the factory site. The circle points denote the samples collected surrounding the factory site. The ellipses individually represent the regions in the factory site and surrounding area grouped by a default 68% confidence interval. Principal component score plot of sampling sites from (**a**) Factory A, (**b**) Factory B.

### 3.2.2. Heavy Metals Classification Using Cluster Analysis (CA)

We used CA to explore commonalities between trace heavy metals and to classify homogeneous groupings. Figure 6 displays the numerous heavy metal clusters in the soil and groundwater of the two factories. For Factory A, the first association, which includes Hg, Cr (VI), and Ni, is labeled as the first cluster, whereas the second, third, and fourth clusters are labeled as Cu, Zn, and Cr, respectively. The association form of the first cluster validates the hypothesis that these heavy metals may be related to plating operations. Factory B has a cluster structure comparable to Factory A except for the kinds of heavy metals, suggesting that these contaminations are mostly from electroplating facilities.

Figure 6c clearly shows that the six elements in Factory A are grouped into two substantial groupings. Cr (VI) and Cr are two heavy metals found in the first cluster. Because of their relatively great distance, Ni and Cu are further categorized into two subclusters. The heavy metals Zn and Hg exist as a distinct subcluster from the

other heavy metals. The heavy metals in Factory B's soil have a similar structure to those in groundwater. Cu was found as a distinct cluster with the greatest distance in soil and groundwater.

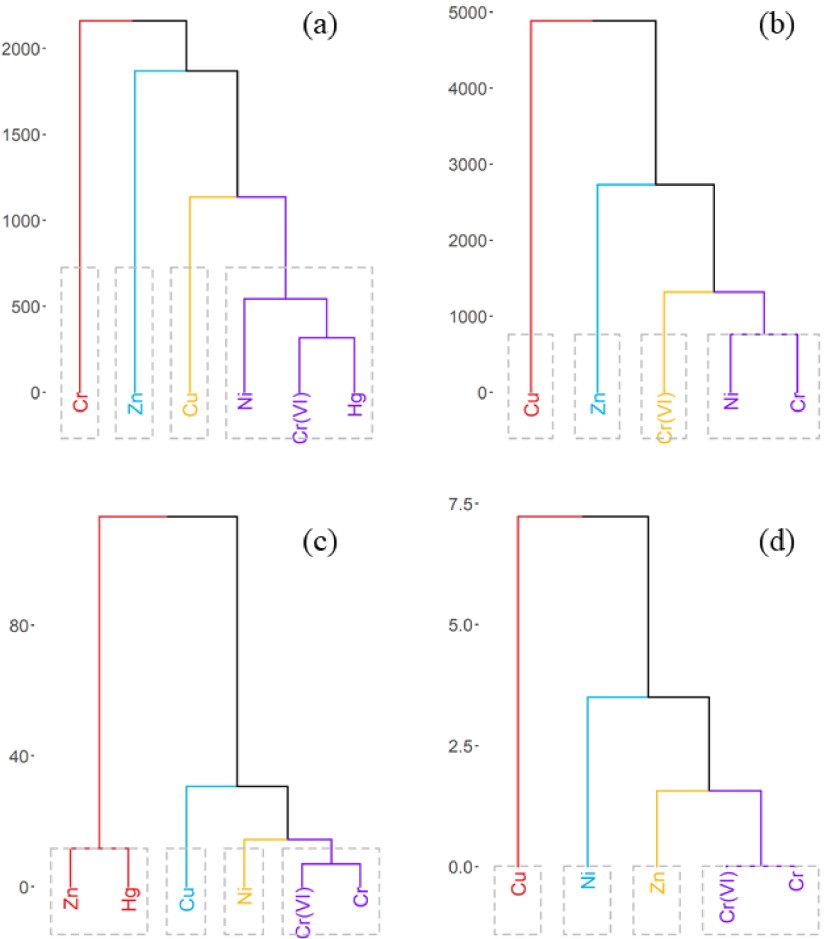

**Figure 6.** Cluster analysis of the heavy metals in soil for (**a**) Factory A, (**b**) Factory B; (**c**) in groundwater for Factory A; (**d**) for Factory B.

3.2.3. Investigation of Heavy Metal in Groundwater and Soil Using Pearson's Correlation Coefficient Analysis

Figure 7 depicts the relationship between heavy metals in soil and groundwater in both electroplating factories and their surrounding areas. We noticed that the Zn and Cu, Cr and Cr (VI), and Cu contents in the soil have extremely substantial positive relationships for Factory A. It can also be shown that there are highly significant positive correlations between Zn and Ni, Zn and Hg, Zn and Cr (VI), and Hg and Cr (VI) ($p \leq 0.05$) in groundwater. This demonstrates that high emissions from Factory A are the predominant cause of heavy metal accumulation in both soil and groundwater. We also found that Cu in groundwater and Cr (VI) exhibit high positive correlations ($p \leq 0.001$), indicating that heavy metal concentrations in soil and groundwater are tightly connected. The more heavy metals there are in the soil, the more heavy metals there are in the groundwater. The Hg content in soil shows no evident link with the other heavy metals, but its correlation with Cu (0.77), Cr (0.44), and Zn (0.45) in groundwater is still significant, showing that Hg comes from the same source as Cd, Cr, and Zn.

As demonstrated in Figure 7b, heavy metals in soil exhibit a substantial connection with Pearson's coefficients larger than 0.7 ($p \leq 0.001$). Similarly, in groundwater, there is a moderately substantial association between Zn and Cr, Zn and Cr (VI), and Cr (VI) and Cr ($p \leq 0.05$). This reveals that heavy metals in the soil and groundwater are derived

from a single source. Most heavy metals in the soil have a negative correlation with those in the groundwater, such as Ni in the groundwater and Cu in the soil ($-0.53$), Ni in the groundwater and Ni in the soil ($-0.51$), and Cu in the groundwater and Cu in the soil ($-0.53$), indicating that the heavy metal in the soil is a source of groundwater.

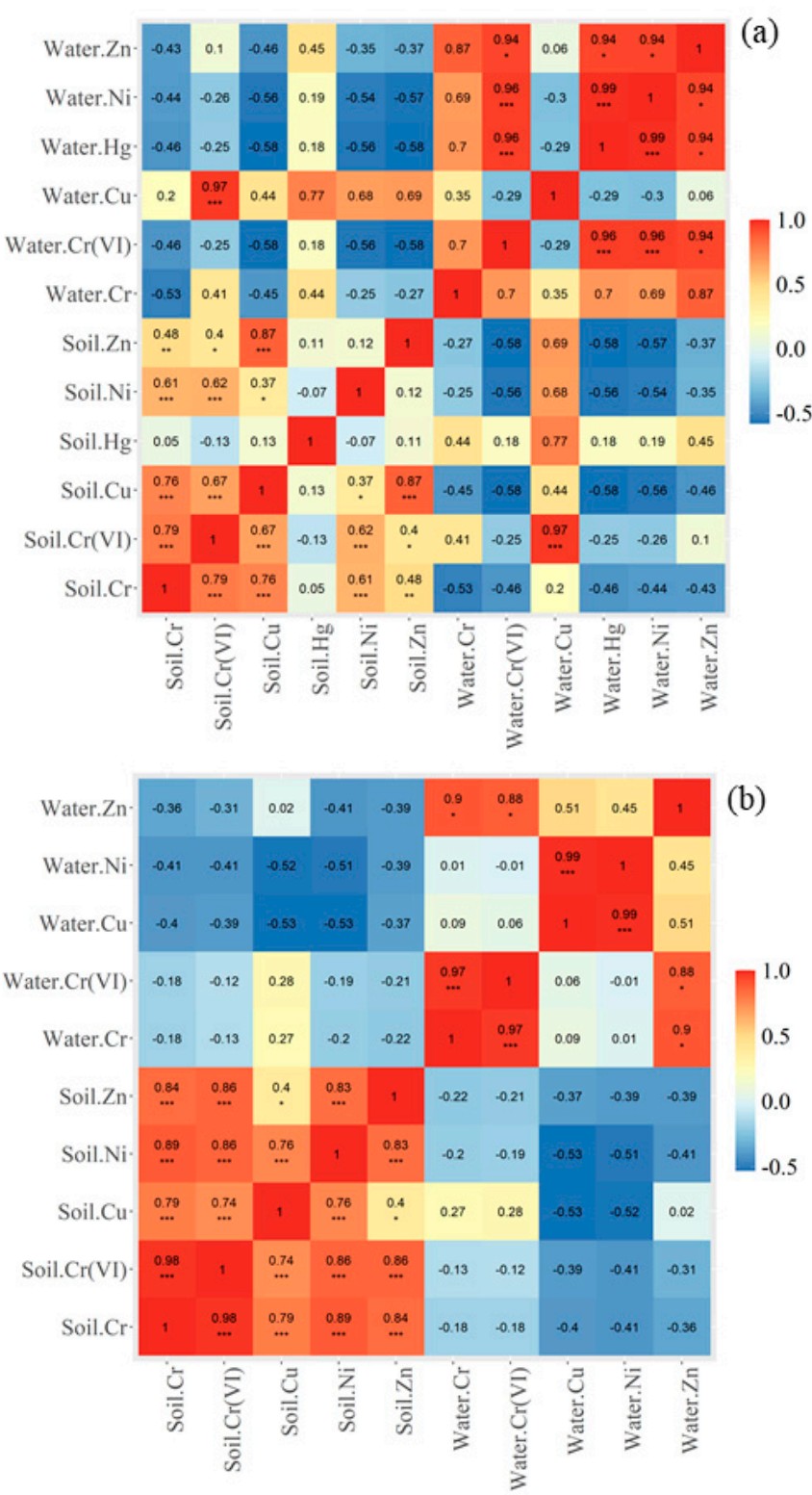

**Figure 7.** Correlation matrix of surveyed heavy metals in soil and groundwater for (**a**) Factory A, (**b**) Factory B. Dotted cell denotes significant correlation (* $p \le 0.5$, ** $p \le 0.01$, *** $p \le 0.001$).

### 3.2.4. The Indicator Ecological Risk Index

For each heavy metal degree, the integral ecological risk of Factory A is ranked as Hg > Cu > Cr (VI) > Ni > Cr > Zn, whereas for Factory B, Cu > Ni > Cr > Zn > Cr (VI). Cu contributes the most to environmental and ecological risk. Because of the comparatively low Cr (VI) content and the relatively low toxic impact of Zn, Cr (VI) and Zn represent a modest ecological danger (10%). According to the distribution of *RI* values for heavy metals (Cr (VI), Cu, Zn, Ni, Hg, Cr) in Figure 8, the *RI* ranges from moderate to considerable risk in Factory A and the adjacent areas. Because Cr, Cu, and Hg in soil contribute significantly to the *RI*, the major high *RI* region occurs at the manufacturing site and the region to the southwest. Solidification/stabilization technology using physical and chemical methods are planned to make the pollutants' diffusion velocity in soil slow down or not migrate [35]. This technology can convert Cr (VI) to Cr, and Cr can be fixed by the soil. Zn, Cu, and Cd are present in both factories. Heavy metals go through adsorption, passivation, and ion exchange treatment during the hydration process due to the unique properties of cement; thus, these heavy metals from around these two factories can eventually stay on the surface of hydrated silicate colloid in the form of hydroxide precipitation or complexes, achieving the effect of eliminating heavy metal harm. In the repair of heavy metal pollutants in groundwater, chemical reduction technology primarily uses chemical medicament's chemical properties to reduce the heavy metals. The application effect of chemical reduction technology is particularly visible in the treatment of heavy metals such as Cr, Ni, and Zn produced by electroplating factories, as the technology not only maintains high removal efficiency but also has a relatively low overall investment cost and has no significant impact on aquifers.

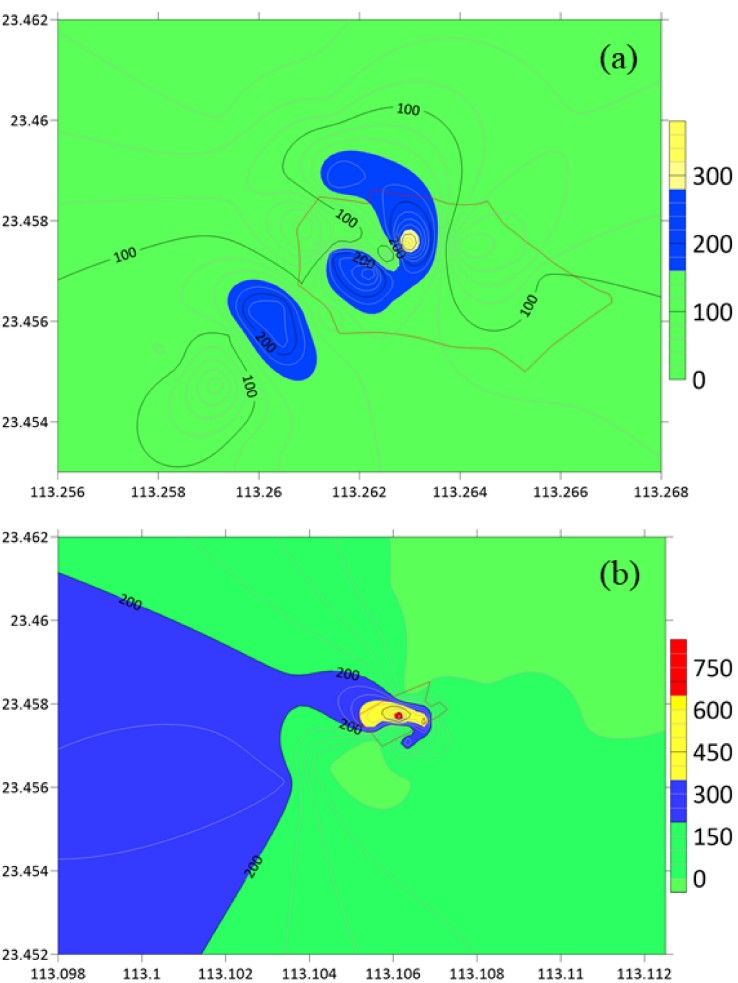

**Figure 8.** Spatial distribution of risk index (*RI*) of heavy metals from (**a**) Factory A, (**b**) Factory B.

The *RI* diminishes as the distance from the production site's center increases. The *RI* of Factory B demonstrates a broad moderate risk zone in the west and southwest. This region is mostly caused by high Cu concentrations in groundwater, demonstrating the harm caused by heavy metal distribution in groundwater. It is highlighted that the majority of Factory B's industrial zones are at high risk, and the *RI* of the workshop at the Factory B center is approximately 800, indicating a high risk level. The in situ chemical oxidation technology is suitable to remove Cu concentration from groundwater. When a particular volume of chemical oxidant is injected into heavy metal-polluted groundwater, these heavy metal contaminants are transformed into compounds with minimal toxicity and mobility by the oxidation process between the chemical oxidant and the heavy metals in groundwater. We employed this method to remove Cu and had good results around these two factories.

## 4. Discussion

### 4.1. Impact on Ecological Environment Considering Heavy Metals in Soil and Groundwater Together

This study investigated the combined influence of heavy metals emitted by electroplating factories on soil and groundwater. Heavy metal concentrations in soil and groundwater around factory sites were many times greater than in surrounding areas, indicating that electroplating factories are the primary cause of the rise in Cr (VI), Cu, Ni, Zn, and Cr concentrations. Only examining the influence of heavy metals in soil would be an underestimation of the ecological impact, as soil mobility is minimal, heavy metals do not spread while adsorbing to soil, and the range of heavy metal contamination in soil is usually fixed [36,37]. Heavy metal pollutants discharged by electroplating factories are eventually deposited in soils and groundwater as a low-solubility compound [38,39]. However, heavy metals migrate widely as a result of groundwater as a medium. For example, Cr (VI) and Ni are very soluble in groundwater, resulting in a wide range of Cr (VI) and Ni concentrations [40,41]. The computed findings of this investigation reveal that polluted soil and groundwater zones do not completely overlap. The polluted region of groundwater is strongly reliant on groundwater flow direction, soil conductivity, and stratum structure. Because of the pollution induced by airborne volatile particles containing heavy metals emitted from chimneys, the polluted area of soil is greatly dependent on wind direction [42,43]. Additionally, the depth of groundwater in Guangdong province is rather shallow, with an average depth of 1.6 m. Heavy metals quickly pollute groundwater, and its discharge might establish a vital channel for pollutants to reach surface water bodies. Furthermore, contaminated groundwater near electroplating sites might pose a long-term harm to residential crops.

### 4.2. Heavy Metal Contamination in Soil as Source

The PCA results demonstrate that distance from the plant locations has a substantial impact on heavy metal pollution. Heavy metals are quickly deposited on the soil's surface and immobilized via adsorption and coagulation [44,45]. When the electroplating factory is operational, the primary source first concentrates on the topsoil. The heavy metal is constantly accumulating, with the largest concentration occurring on the soil's surface. When a facility, such as Factory B in our research, is shut down, it is still difficult to remove the heavy metals in the soil, and some of them travel downward due to gravity and leaching [46,47].

Several earlier investigations have indicated that soil adsorption plays an important role in heavy metal sequestration [48]. Land use [49], organic matter in agricultural and forest soils [50], and clay minerals or metal oxides in urban soils and sediments are all elements that may influence heavy metal diffusion [51,52]. Finer soil particles have a greater potential to absorb heavy metals. According to Huang et al. [53], soil particle size has a substantial impact on the environmental behavior of heavy metals in soil. Greater water conductivity around Factory B, as well as broader areas of crops and forest surrounding Factory B, resulted in the easy penetration of heavy metals to considerable depths in this research. As shown in Figure 4h–k, the concentration of heavy metals in Factory B's

groundwater is substantially greater, and the maximum concentration exists at a deeper level than in Factory A.

*4.3. Influence of Heavy Metals in Soil and Groundwater on Plants*

Heavy metal contamination in soil and water is becoming a significant concern to plants. The nature of the heavy metals in fly ash is mostly determined by particle size and mineralogy [54]. The majority of ash is disposed of through effluent outputs that enter local water circulation [55]. If the soil absorbs the fly ash, a considerable portion of the ash leach percolates into the groundwater level. Water movement, such as precipitation with soluble components, evidently has an impact on the surrounding soil and groundwater. Soluble heavy metals slowly penetrate and pollute groundwater aquifers near electroplating plants [56]. The release of heavy metals from ash in topsoil increases as pH decreases [57].

Plants use a variety of strategies to maintain physiological concentrations of essential metal ions while minimizing exposure to unneeded heavy metals [58]. Heavy metals enter plant shoots and roots via the plasma membrane from soil and groundwater [59]. Metal ions are acquired by the roots, shoots, and leaves and, subsequently, are transported and distributed throughout the plant; the management of their cytosolic concentrations is crucial for plant growth [60]. This study evaluates the influence of heavy metals on plants to some extent by taking metals in both soil and groundwater into account, but it does not directly address the impact on different plant species. Filling such information gaps can assist us in better understanding and forecasting the impact of heavy metals in different plant and environmental evaluations.

**5. Conclusions**

The research took into account heavy metals in both soil and groundwater. Heavy metal pollutants discharged by electroplating factories are deposited in soils and groundwater as a low-solubility compound. The heavy metal migrates widely as a result of the use of groundwater as a medium. Contaminated soil and its discharge readily pollute groundwater. The polluted groundwater in the vicinity of the electroplating regions may pose a long-term harm to residential crops. The heavy metal concentrations were compared between Factory A, which is in operation, and Factory B, which is no longer in operation, in order to analyze the heavy metal concentrations and associated ecological risks.

The PCA results demonstrated that the heavy metal pollution inside Factory A had a substantially greater impact than in the surrounding areas. The data show that concentration fell within Factory B, but was increasing outside of it. The concentrations in these two places are adversely connected, and the impact of contaminated regions is approaching. The cluster analysis demonstrated that these pollutions were mostly the result of electroplating facilities. According to Pearson's correlation coefficient research, even when the electroplating facility is closed and the production is stopped, heavy metal contamination continues to spread and penetrate, and more heavy metals are released from the soil to the groundwater as time passes. The heavy metal is constantly accumulating, with the largest concentration occurring on the soil's surface when the factory is in operation. However, removing heavy metals from the soil remains challenging. The concentration of heavy metals in Factory B's groundwater was substantially higher, and the maximum concentration resides far deeper than in Factory A. Soil and groundwater remediation is generally referred to as the construction that removes contaminants from the sites. Future research should thoroughly investigate the impact of various remediation procedures, prediction modeling, and assessment on the sustainability of these enterprises.

**Author Contributions:** Conceptualization, H.F. and J.Q.; methodology, H.F.; software, W.Y.; validation, D.X. and X.L.; investigation, X.W. and J.Z. (Jingwen Zeng); data curation, Y.Z.; writing—review and editing, J.Z. (Jianting Zhu); writing-original draft, H.F.; funding acquisition, Y.L. and J.Q.; visualization, Y.S. All authors have read and agreed to the published version of the manuscript.

**Funding:** This study was supported by the National Key Research and Development Program of China (2018YFC1800304), the Key-Area Research and Development Program of Guangdong Province (2019B110205002), and the Youth Innovation Fund of Eco-environment Remediation Research Center, SCIES (hx_202109_002). Funding was from Science and Technology Projects in Guangzhou (202002030276).

**Institutional Review Board Statement:** Not applicable.

**Informed Consent Statement:** Not applicable.

**Acknowledgments:** We highly appreciate the valuable comments from anonymous reviewers that greatly improved our manuscript.

**Conflicts of Interest:** The authors declare no conflict of interest.

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
