# Peer review of "Improvement of Ecological Risk Considering Heavy Metal in Soil and Groundwater Surrounding Electroplating Factories"

_processes, doi:10.3390/pr10071267_

Round 1

Reviewer 1 Report

A rather interesting manuscript is presented. Heavy metals in groundwater and soil around two different factories were analysed. The manuscript has been generally well written and structured. I just have some suggestions for revision of the manuscript before publication in the Journal of Processes given below:

·        There is some missing information related to both factories (A and B) that can be affected the results such as

1.      Time of operation of B and even for A so far.

2.      The amount of loading per year for both factories.

3.      The main electroplating metals, the procedure for electroplating and the process of management of their waste which authors must briefly add in the manuscript.

·        The marks of A, B, C, and D are missing in the Figure1

Author Response

Reviewer 1:

Comment 1: A rather interesting manuscript is presented. Heavy metals in groundwater and soil around two different factories were analyzed. The manuscript has been generally well written and structured. I just have some suggestions for revision of the manuscript before publication in the Journal of Processes given below:

Response 1: Thanks for the comments. We have revised the manuscript according to your editorial suggestions. We have also very carefully addressed the comments, which were highlighted in the revised manuscript.

Comment 2: There is some missing information related to both factories (A and B) that can be affected the results such as 1. Time of operation of B and even for A so far. 2. The amount of loading per year for both factories.

Response 2: Factory A was constructed in 2000 and its primary manufacturing job is to apply a layer of chromium metal coating to the surface of the plated components, giving the surface of the plated parts a certain wear resistance and a bright silver-white appearance. About 10 million electroplating parts are produced each year. Factory B was established in 1986 and its contaminated source is primarily electroplating wastewater. It produces about 9 million electroplating parts per year. These parts of the introduction were added at Line 117-120, 122-123.

Comment 3: The main electroplating metals, the procedure for electroplating and the process of management of their waste which authors must briefly add in the manuscript. The marks of A, B, C, and D are missing in the Figure1.

Response 3: These two factories' manufacturing processes were briefly described at Lines 117-120, 124-126. The marks of A, B, C, and D were added to the revised manuscript.

Reviewer 2 Report

Hong Fang et al, reported a research paper titled " Improvement of ecological risk considering heavy metal in soil and groundwater surrounding electroplating factories.

Authors studied heavy metal contamination in ground water and soil near electroplating factories A and B. Overall the article lacking scientific novelty, however the article is well written. It is obvious that the soil and water near industrial area (including electroplating factories) contaminated by metals and other pollutants from the industries. If author develop and discuss any remediation technology in this article that will increase the impact of the article in the scientific community.

Author Response

Reviewer 2:

Comment 1: Hong Fang et al, reported a research paper titled " Improvement of ecological risk considering heavy metal in soil and groundwater surrounding electroplating factories. Authors studied heavy metal contamination in ground water and soil near electroplating factories A and B. Overall the article lacking scientific novelty, however the article is well written.

Response 1: Thanks for the comments. In the revision, we have carefully addressed the comments as explained below.

Comment 2: It is obvious that the soil and water near industrial area (including electroplating factories) contaminated by metals and other pollutants from the industries. If author develop and discuss any remediation technology in this article that will increase the impact of the article in the scientific community.

Response 2: The discussion of remediation technology has been added to the updated manuscript. Soil and groundwater contamination remediation technology can be divided into two types based on cleanup procedures: ectopic and in situ. Ectopic repair is mostly an extraction method. Ectopic restoration has the advantage of a short repair cycle and is suitable for areas with relatively concentrated pollution sources. The remediation technology is suitable for treating large polluted areas at a low cost and with minor environmental impact. Our research focuses primarily on in situ soil and groundwater remediation to remove heavy metals from the soil and groundwater. This part of the discussion has been incorporated at Line 422-486.

Reviewer 3 Report

None

Author Response

Reviewer 3:

Comment: English language and style are fine/minor spell check required.

Response: Thanks for the comments. In the revised manuscript, we have carefully checked the English language and make revisions.

Reviewer 4 Report

The revised version of this manuscript seems to be an intersting work.

(2 refs. published in 2022 and 3 refs. published in 2021) More recent refernces published in 2021 and 2022 are recommended to be cited in the manuscript. 

Author Response

Reviewer:

Comment: The revised version of this manuscript seems to be an interesting work. (2 refs. published in 2022 and 3 refs. published in 2021) More recent references published in 2021 and 2022 are recommended to be cited in the manuscript.

Response: Thanks for the comments and suggestions. More references published in 2021 and 2022 have been cited in the revised manuscript.

Round 2

Reviewer 2 Report

Authors discussed about various remediation technologies available in the literature. Unfortunately authors haven't applied any remediation methods (didn't have any experimental data) to treat heavy metal contamination from soil and water near the electroplating methods. Instead of reffering to the existing remediation methods in the literature, reviewer suggest authors apply the suitable remediation methods and have experimental data and discuss the results so that readers know how much effective the used remediation method in treating heavy metal contamination at the elctroplating industrial area.

Author Response

Reviewer 2:

Comment: Authors discussed about various remediation technologies available in the literature. Unfortunately, authors haven't applied any remediation methods (didn't have any experimental data) to treat heavy metal contamination from soil and water near the electroplating methods. Instead of referring to the existing remediation methods in the literature, reviewer suggest authors apply the suitable remediation methods and have experimental data and discuss the results so that readers know how much effective the used remediation method in treating heavy metal contamination at the electroplating industrial area.

Response: Groundwater contamination is heavily influenced by groundwater flow direction and is mostly disseminated in the groundwater's downstream area. The area contaminated by soil is heavily influenced by wind direction. These two areas are not interchangeable. Most previous studies calculated the ecological risk index of soil or groundwater individually and did not account for the influence of both. This study is unique in that it comprehensively evaluates both soil and groundwater contamination in and around the electroplating factory. In the non-overlapping area, the ecological risk index of soil and groundwater is computed individually, and in the overlapping area, the greater ecological risk index of soil and groundwater is employed. The soil and groundwater risk indices were thoroughly compared, taking into account the continuous discharge of pollution sources from the electroplating facilities and the cessation of discharge of pollution sources following the closure of the electroplating factory. (Lines 100-111)

The purpose of this research is not to develop new remediation technology. Based on the ecological risk index calculated from the improved method, this study recommends several ecological restoration approaches suitable for deployment in various regions.
